# Integration of UAV, Sentinel-1, and Sentinel-2 Data for Mangrove Plantation Aboveground Biomass Monitoring in Senegal

**José Antonio Navarro [1,2,](ID), Nur Algeet [1], Alfredo Fernández-Landa [1] (ID), Jessica Esteban [3], Pablo Rodríguez-Noriega [1] and María Luz Guillén-Climent [1]**

[1]    Agresta Soc. Coop., 28012 Madrid, Spain; nalgeet@agresta.org (N.A.); afernandez@agresta.org (A.F.-L.); prodriguez@agresta.org (P.R.-N.); mguillen@agresta.org (M.L.G.-C.)
[2]    MONTES (School of Forest Engineering and Natural Environment), Universidad Politécnica de Madrid, 28040 Madrid, Spain
[3]    Departamento de Topografía y Geomática, ETSI Caminos, Canales y Puertos, Universidad Politécnica de Madrid, 28040 Madrid, Spain; jesteban@upm.es
*    Correspondence: janavarro@agresta.org

**Abstract:** Due to the increasing importance of mangroves in climate change mitigation projects, more accurate and cost-effective aboveground biomass (AGB) monitoring methods are required. However, field measurements of AGB may be a challenge because of their remote location and the difficulty to walk in these areas. This study is based on the Livelihoods Fund Oceanium project that monitors 10,000 ha of mangrove plantations. In a first step, the possibility of replacing traditional field measurements of sample plots in a young mangrove plantation by a semiautomatic processing of UAV-based photogrammetric point clouds was assessed. In a second step, Sentinel-1 radar and Sentinel-2 optical imagery were used as auxiliary information to estimate AGB and its variance for the entire study area under a model-assisted framework. AGB was measured using UAV imagery in a total of 95 sample plots. UAV plot data was used in combination with non-parametric support vector regression (SVR) models for the estimation of the study area AGB using model-assisted estimators. Purely UAV-based AGB estimates and their associated standard error (SE) were compared with model-assisted estimates using (1) Sentinel-1, (2) Sentinel-2, and (3) a combination of Sentinel-1 and Sentinel-2 data as auxiliary information. The validation of the UAV-based individual tree height and crown diameter measurements showed a root mean square error (RMSE) of 0.21 m and 0.32 m, respectively. Relative efficiency of the three model-assisted scenarios ranged between 1.61 and 2.15. Although all SVR models improved the efficiency of the monitoring over UAV-based estimates, the best results were achieved when a combination of Sentinel-1 and Sentinel-2 data was used. Results indicated that the methodology used in this research can provide accurate and cost-effective estimates of AGB in young mangrove plantations.

**Keywords:** digital aerial photogrammetry; SAR; model-assisted; biomass estimation; Copernicus; unmanned aerial vehicles

## 1. Introduction

Mangroves are highly productive ecosystems and are able to sequester and store large amounts of carbon [1–3]. They also play a key role in production of timber and non-timber forest products, shoreline protection, providing fishing areas, or filtering water pollution [1,4]. For these reasons mangrove ecosystems are highly interesting zones for climate mitigation and adaptation projects [5].

In the last few years, the attention to afforestation and reforestation projects as well conservation programs, such as Reducing Emissions from Deforestation and Forest Degradation Plus (REDD+), has increased. These programs require accurate estimations of biomass and carbon stocks in vegetation and soils to monitor changes in extent, carbon emissions, and sequestration rates. The use of new technologies based in remote sensing can improve the accuracy of monitoring and enhance our understanding of the changes in forested mangrove areas [6].

Traditional inventory data collection methods may be accurate and offer detailed information on the composition and structure of forests [7]. However, this task can be inefficient or time-consuming [8] in remote or hard-to-reach locations and difficult-to-work areas (i.e., mangroves). Monitoring mangrove forests is also arduous because of their large extent [9], thus remote sensing data has been widely used for this purpose. The kind of remote sensing platform used depends on the scale and the goal of the research [10]. Low- and medium-resolution space-borne sensors availability is generally free of charges or cheaper than airborne sensors and they offer larger coverage area while airborne and unmanned aerial vehicles (UAV) sensors have much more spatial resolution but limited autonomy, which can result in a higher cost per hectare. Giving that soil is the main carbon pool in this type of forests [3,11] most studies have focused on investigating changes in mangrove land cover [12] since soil carbon is relatively stable [11].

Many studies have monitored the mangrove forests coverage using space-borne imagery, from low-resolution sensors, such as MODIS [13], or medium-resolution satellite imagery, such as LANDSAT [14], to very-high resolution (VHR) imagery from WorldView-2 [15]. Recent studies have analyzed the vertical structure of mangrove forests from space-borne and airborne observations. Simard et al. [16] and Fatoyinbo et al. [17] used the Shuttle Radar Topography Mission (SRTM) for mangroves canopy height estimation, while Polarimetric Synthetic Aperture Radar Interferometry (Pol-InSAR) was applied to data collected from the TanDEM-X InSAR (TDX) by Lee and Fatoyinbo [18]; Lee et al. [19], and Fatoyimbo et al. [20]. VHR satellite stereophotogrammetry has also been used to create canopy height models (CHM) [6]. On the other hand, the airborne laser scanner (ALS) also provides elevation data to estimate canopy heights and to calibrate and validate estimations from space-borne remote sensing sensors [6,9,20].

Within the satellite remote sensing techniques, synthetic aperture radar (SAR) sensors can be more effective for monitoring forest biomass since they are independent of cloud conditions [21] and can penetrate the canopy [22,23]. SAR sensors use different wavelengths which are able to penetrate the forest in different ways [23]. The X- band and C-band are sensitive to leaves and needles [24]. These bands are suitable for monitoring young growth stages of mangrove forests or plantations [25]. The launch of Sentinel-1A and Sentinel-1B enables very frequent SAR data acquisitions under a free data policy. Sentinel-1 provides SAR images with a high geometric resolution (5 m $\times$ 20 m on the ground) with HH+HV or VV+VH polarizations in the C-band [26]. Nevertheless, C-band backscatter saturation levels are typically low in mangrove biomass estimations (50–70 Mg ha$^{-1}$) [27,28]. Some studies have demonstrated that the integration of SAR and optical sensors data improves forest biomass estimates since optical data contributes to offset the saturation effect [29,30]. Thus, the opportunities for mangrove biomass monitoring have improved with the subsequent launch of Sentinel-2 (multispectral) satellites of the European Commission's Copernicus program.

Models relating observations of forest attributes measured on field plots and remotely-sensed data for the same plots are often used when plot-based estimates are not sufficiently precise or there are not enough field plots available [31]. Model-based inference is based solely on assumptions of the model [32]. Therefore, under model-based frameworks estimators may be both biased and imprecise depending on the goodness of the model [32]. On the other hand, the use of models in the context of design-based inference does not have this problem and models may be used to enhance the variance [33]. In this way, an inadequately specified model using design-based inference through model-assisted estimation will not lead to bias estimators [32]. Model-assisted frameworks have been extensively used in large-area aboveground biomass (AGB) monitoring [31,33–35].

Non-parametric models have been widely used for AGB estimation [29,30,36–38]. The use of machine learning algorithms, such as k-Nearest Neighbor (k-NN), back propagation neural networks (BPNN), multilayer perceptron neural network (MLPNN), random forest (RF), or support vector regression (SVR), have been extended due to their ability to model relatively easy complex non-lineal relationships between the variables and to process large dataset efficiently [39]. Although parametric models have been more frequently used in connection with model-assisted estimation, non-parametric models have also been employed [40,41].

During the past few years, dense image matching (DIM) has reached great importance in digital surface models (DSM) generation due to the improvements in hardware and photogrammetric algorithms, such as structure-from-motion (SfM) [42,43]. Three-dimensional (3D) ALS-like point clouds may be produced by photogrammetric matching of digital aerial images [44–46]. However, DIM-based point clouds only provide information at the top surface, therefore, an accurate bare-earth digital elevation model (DEM) for estimating canopy height and structure is essential [47]. However, DEMs may be produced by photogrammetry without any support from other sensors in open canopy forests [48]. Furthermore, recent studies have shown the application of UAVs in forest variables estimation [49–54]. One of the main advantages of DIM-based point clouds generated from UAV imagery is the capacity to detail the vegetation at the centimeter level [55]. VHR imagery allows for individual tree crown (ITC) extraction from the DIM-derived canopy models [56] and for measuring parameters like individual tree height or crown surface. Such measurements are very useful as they are good estimators of other interest variables as, inter alia, diameter at breast height, volume, AGB, or tree growth [54,57]. Thus, these facts coupled with the low operational cost of UAVs [58] has resulted in UAVs being used as a popular alternative in ecosystems for surveying and mapping [59].

Particularly, only a few studies have researched the application of UAVs to the mangrove ecosystems [60–62]. Nevertheless, UAVs may be a practical solution in remote areas since they allow us to develop rapid and cost-effective surveying forest attributes [52,61,63,64]. Using UAVs also provides an advantage over other remote sensing systems due to the possibility to plan imagery capture during low sea tide. Although digital photogrammetry from UAVs leads to good estimations of mangrove forests parameters, it is costly for wall-to-wall large-scale forest inventories. Instead, UAVs may be used in assessing tree variables at the plot-scale. To our knowledge, only Mayr et al. [65] have researched the use of photogrammetric point clouds from UAVs to delineate tree crowns in separate plots.

This study purposes a novel technique to quantify AGB in large areas of young reforested mangroves replacing traditional field sampling methods by photogrammetric point cloud-based measurements and using wall-to-wall Sentinel-1 and Sentinel-2 data as auxiliary information. The aims of this study were (1) to evaluate the performance of low-cost UAV-derived photogrammetric point clouds for the measurement of individual tree heights and crown diameters, (2) to investigate the usability of wall-to-wall Sentinel-1 and Sentinel-2 data as auxiliary information for estimating the AGB using a probability sampling design, and (3) to compare the AGB estimates and their precisions for the different satellite data.

## 2. Materials and Methods

### 2.1. Study Area

This study was conducted in the mangrove forest of Senegal in the Sine Saloum and Casamance Deltas (12°20′–14°10′N; 15°24′–16°47′W) located in the west coast of the country (Figure 1). The study area is located in a mangrove restoration project with a total area of 10,415.12 ha that was planted between 2009 and 2012 (1550.05 in 2009; 4285.14 in 2010; 3337.47 in 2011 and 1242.46 in 2012). The project area consists of 2657 planted parcels, scattered in the deltas, with a mean area of 3.92 ha. The species planted is *Rhizophora mangle* L. with a mean planting density of 5000 trees/ha [66].

The Sine Saloum delta is in the Sudanese climate domain with annual precipitation ranging from 450–920 mm, while the parcels located in the Casamance area are in the Sudanese-Guinean and sub-Guinean climate domains where annual precipitation ranges between 800–1700 mm [67]. Average air temperature ranges from 26 °C to 29.7 °C in the Casamance area and between 27.2 °C and 30 °C in the case of Sine Saloum [68]. The monsoonal rainy season is a result of the St. Helen High and lasts from June to September [69].

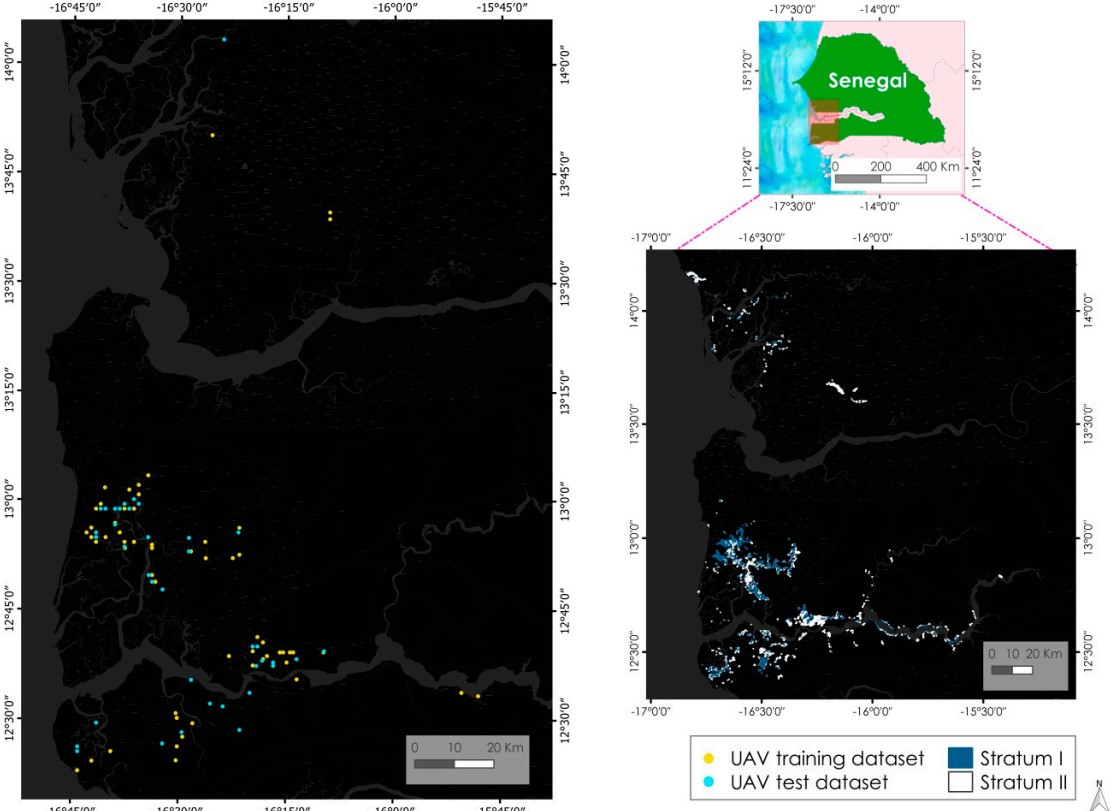

**Figure 1.** Overview of the study area on the west coast of Senegal, stratification, and sampling design.

## 2.2. Satellite Data. Acquisition, and Preprocessing

Sentinel-1 dual-polarized images in Interferometric Wide Swath (IW, 250 km swath width) and Sentinel-2 images were acquired, from the European Space Agency (ESA) Sentinel science hub (https://scihub.copernicus.eu/) (Table 1), to provide AGB estimations. Additionally, Sentinel-2 data was used to stratify the study area.

**Table 1.** Remotely-sensed data acquisition.

| Sensor | Application | Acquisition Period | Processing Levels | Bands |
|--------|-------------|--------------------|-------------------|-------|
| S2 | Stratification | February 2017 March 2017 (7 scenes) | Level-1C | B2, B3, B4, B5, B6, B7, B8A, B9, B11, B12 |
| | AGB prediction | July 2017 August 2017 (6 scenes) | | |
| S1 | AGB prediction | July 2017 September 2017 (6 scenes) | Level-1 GRD | C-band (VH polarization) |

S2 = Sentinel-2; S1 = Sentinel-1; GRD = ground-range detected.

The study area region is covered by two Sentinel-1 scenes and for each one, three Standard Level 1 Products GRD (ground-range detected) were acquired in ascending mode in the same period of time within which the UAV-based inventory was carried out and taking into account the sea tide level at the satellite acquisition time over the study area. Sentinel-1 products processing workflow consisted of four steps achieved in SNAP 6.0 software [70]: (i) radiometric calibration (output was sigma0 band), (ii) terrain-correction, based on SRTM digital elevation model (three-second resolution), (iii) single-product speckle filtering based on a three pixel size Lee filter, and (iv) a linear conversion to dB. The outputs were backscatter images at 20 m resolution. In this study, only VH polarization images were included in the modeling scheme since it has been shown to be more efficient than VV and HH for the AGB estimation because it is less influenced by soil moisture [71]. The three date data were averaged to generate a mean VH polarization image.

For our study purposes, we used the spectral bands with 10 m and 20 m resolutions, while bands at 60 m were excluded from the analysis. The Sentinel-2A Level 1-C Top-of-Atmosphere (TOA) reflectance images were processed to Level-2A bottom-of-atmosphere (BOA) values using the freely available SNAP toolbox [70] and the associated Sen2Cor plug-in [72]. Sentinel-2 bands were resampled to match the 20 m spatial resolution of the Sentinel-1 VH polarized backscatter. Ten different vegetation indices were generated from multispectral Sentinel-2 data (Table 2).

**Table 2.** Sentinel-2 imagery data bands and vegetation indices used in this study.

| Predictor Variable | Band/Index | Definition |
|---|---|---|
| **Multispectral bands** | B2 | Blue, 490 nm |
| | B3 | Green, 560 nm |
| | B4 | Red, 665 nm |
| | B5 | Red edge, 705 nm |
| | B6 | Red edge, 749 nm |
| | B7 | Red edge, 783 nm |
| | B8 | Near Infrared (NIR), 842 nm |
| | B8A | Near Infrared (NIR), 865 nm |
| | B9 | Water vapor, 945 nm |
| | B11 | Short-wavelength infrared (SWIR-1), 1610 nm |
| | B12 | Short-wavelength infrared (SWIR-2), 2190 nm |
| **Vegetation indices** | NDVI1 | $(B8 - B4)/(B8 + B4)$ |
| | NDVI2 | $(B8A - B4)/(B8A + B4)$ |
| | NDI45 | $(B5 - B4)/(B5 + B4)$, |
| | SAVI | $(B8 - B4)/(B8 + B4 + L) * (1.0 + L)$ <br> $L = 0.5$ |
| | TCARI | $3 * [(B5 - B4) - 0.2 * (B5 - B3) * (B5/B4)]$ |
| | OSAVI | $(1.16) * (B8 - B4)/(B8 + B4 + 0.16)$ |
| | MCARI | $[(B5 - B4) - 0.2 (B5 - B3)] * (B5/B4)$ |
| | GNDVI | $(B8 - B3)/(B8 + B3)$ |
| | PSSRa | $B8/B4$ |
| | IRECI | $(B8 - B4)/(B5/B6)$ |

NDVI = Normalized Difference Vegetation Index; SAVI = Soil Adjusted Vegetation Index; TCARI = Transformed Chlorophyll Absorption Ratio Index; OSAVI = Optimized Soil Adjusted Vegetation Index; MCARI = Modified Chlorophyll Absorption in Reflectance Index; GNDVI = Green Normalized Difference Vegetation Index; PSSRa = Simple Ratio 800/680 Pigment Specific Simple Ratio (Cholophyll a); IRECI = Inverted Red-Edge ChlorophyllIndex.

For all survey plots, bands, vegetation indices from the Sentinel-2 Level 2-A and backscatter mean values from the Sentinel-1 products were computed using the *Extract* function in the *Raster* package [73] within the R software environment [74].

A flowchart showing the general research framework for estimating the AGB of mangrove plantations used in this study is presented in Figure 2.

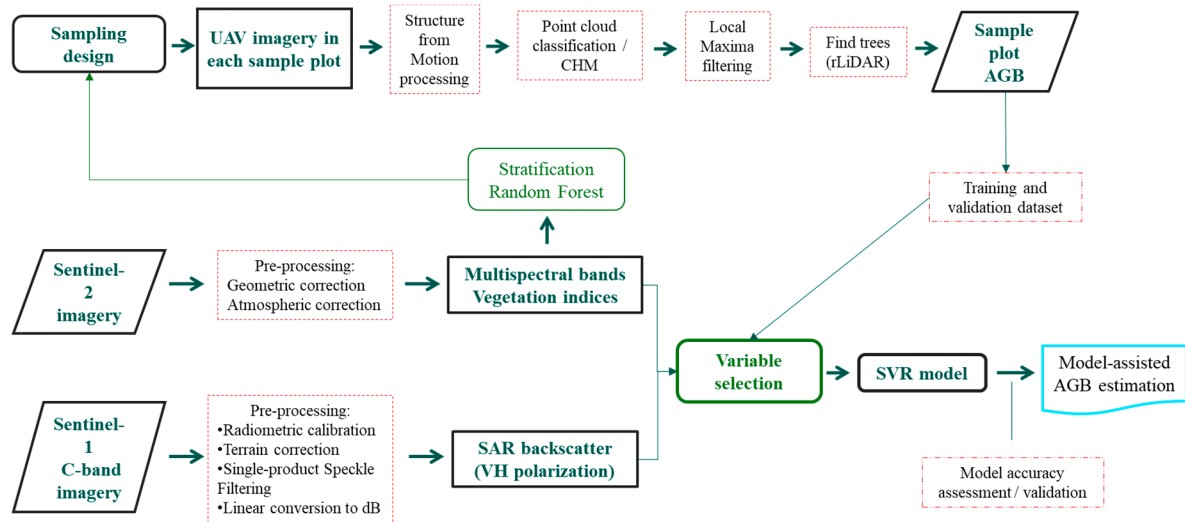

**Figure 2.** General methodology workflow used for AGB estimation integrating the Sentinel SAR and multispectral data. UAV-derived imagery was used for sampling plot measuring.

### 2.3. Stratification and Sampling Design

A total of seven Sentinel-2 images were initially used as auxiliary information to generate a wall-to-wall stratification of the mangrove plantations in the study area, which aids to encompass the full range of available AGB. Two main strata were defined to allocate efficiently a stratified systematic sample of UAV-based plots in each of them, (i) one with very low AGB densities due to high plantation mortality and low plantation development stage (Stratum I) and (ii) the second stratum with higher AGB densities and therefore, lower mortality rates and higher tree cover and plantation development stage (Stratum II).

The random forest (RF) classifier [75] was selected to perform the stratification classification and it was implemented using the R-Package *Random Forest: Breiman and Cutler's Random Forests for classification and regression* [76]. Random forests improve classification accuracy by growing an ensemble of classification trees and letting them vote on the classification decision. For the model training, regions of interest (ROIs) of the two strata were manually defined based on field information and observations of canopy cover over high resolution images from Google Earth.

Two RF models were fitted, one for the classification of 9th March 2017 (western zone of the study area) images and one for classifying 24 February 2017 images (eastern zone). A total of 20 variables were used as predictors in the image classification procedure: 10 Sentinel-2 spectral bands and 10 Sentinel-2 vegetation indices (Table 2). A random forest variable selection algorithm *VSURF* [77] was applied to reduce the number of predictor variables and to improve the performance of the Random Forest models (Table 3).

**Table 3.** Predictor variables from Sentinel-2 imagery data used in random forest classification and variables finally selected by VSURF.

| Images Date | RF Predictor Variables | VSURF Selected Variables |
|:---:|:---:|:---:|
| 9 March 2017 | B2, B3, B4, B5, B6, B7, B8, B8A, B9, B12, NDVI1, NDVI2, NDI45, SAVI, TCARI, OSAVI, MCARI, GNDVI, PSSRa, IRECI | B3, B12, OSAVI, NDVI2, NDI45, B9, B8 |
| 24 February 2017 | B2, B3, B4, B5, B6, B7, B8, B8A, B9, B12, NDVI1, NDVI2, NDI45, SAVI, TCARI, OSAVI, MCARI, GNDVI, PSSRa, IRECI | NDVI2, B3, B9, B12 |

The areas of the two strata in the study region were 6927.03 ha for Stratum I and 3488.09 ha for Stratum II. A stratified systematic sample of UAV-based plots was designed for the two pre-defined strata. The sample size was adapted to suit the time available to carry out the inventory, so the sample consisted of 95 circular plots with a radius of 10 m and an area of 314.16 m$^2$ plots. The dataset was randomly divided into 60% training and 40% validation data samples (57 and 38 sample plots respectively). Based on a target of equal allocation of $n_h$ = 50 plots per stratum, the systematic sample was distributed on a grid of 1200 m by 1200 m and 850 by 850 m in Stratum I and II, respectively. The spacing of the grid was determined as:

$$l_h = \sqrt{\frac{A_h}{n_h}}\,, \qquad\qquad (1)$$

where $l_h$ is the spacing of the grid for the stratum $h$, $A_h$ is the size of the stratum $h$ (m$^2$), and $n_h$ is the initial number of sample plots for the stratum $h$.

*2.4. Sampling Data Collection and Processing*

Aerial and field measurement campaigns were done simultaneously during the months of July and August 2017 in low tide conditions. UAV imagery was acquired in 95 flights (i.e., one per sample plot) using a commercial compact Parrot Bebop 2 quadricopter (Parrot SA, Paris, France). Pix4dCapture software (Pix4D SA, Lausanne, Switzerland) was used to design and guide each mission flight. In order to achieve a better 3-D representation of the plots, a circular mission option was chosen since it is recommended for small areas and 3D model outputs (e.g., point clouds) at an altitude of 25 m above ground. This kind of mission ensures that the images are taken from all angles around a point of interest with the required overlap for photogrammetric processing [78]. The oblique imagery provides a more detailed characterization of the sample trees [79].

Tree height and crown diameters were measured in the field on 100 different trees (between one and three trees per plot) to compare and validate field and UAV-based measurements. Before each UAV flight, colored disks were placed next to the measured trees to locate them in the point clouds. Tree heights and crown diameters measured over the photogrammetric point cloud were added to the trees database.

Each of the 95 flights was processed separately for DIM point clouds and orthomosaics generation using the photogrammetry software Pix4Dmapper (Pix4D SA, Lausanne, Switzerland) [80]. Pix4Dmapper uses proprietary algorithms based on computer vision structure-from-motion (SfM) and stereo-matching algorithms to align images and build a georeferenced sparse point cloud. For images taken with the Bebop 2 drone, the sky is automatically removed in the image alignment phase in Pix4D. After this step the point cloud is densified using multi-view stereo-reconstruction algorithms. For this study, a half size image scale was used for the point clouds densification as it is recommended by the software developers [78]. As no additional ground control points (GCP) were collected for enhancing accuracy, only the GPS coordinates of the tagged images were used in this process. Hence, coordinates of sample plots were determined by the UAV GPS/INS system and the photogrammetric reconstruction.

Bare-earth was extracted from the point clouds and the height above the ground was computed for each point using the *Lasground* tool in Lastools [81]. The algorithm parameters were also fine-tuned for an optimum result (step was 2 m, bulge was 1 m, spike was 0.01 m and standard deviation was 10 m). The point clouds were clipped to the spatial extent of the sample plots (i.e., circular with a radius of 10 m). Ground points were used to generate gridded DTMs with a resolution of 0.5 × 0.5 m and the vegetated above ground points were used to create 0.1 m DSMs. After this, a 0.1 m resolution CHM was generated for each sample plot by subtracting bare earth heights from the DSM heights. FUSION software [82] was used for DTM and CHM creation.

Heights and positions of individual trees inside the plot were determined using local maxima filters based on a locally-variable window size [83–85]. This algorithm identifies the highest point within a variable window. For this, the filter moves the window over the CHM and uses a circular window to determine if the center pixel is a local maximum by comparing this pixel with the surroundings pixels within the window. Window size depends on tree height by referring to a predefined height- crown equation. In order to achieve the best results and taking in account that tree sizes are different in each plot, various windows sizes were tested using the *CanopyMaxima* function of FUSION. Afterwards, a manual debug of the results was carried out using the orthomosaics derived from UAV data as reference to ensure the measurement of all trees in plots. Finally, tree crown surfaces were delineated computing the rLidar package [86] within the R software environment [74]. The tree crown diameter was calculated as the diameter of the circle with equivalent area:

$$cd = \sqrt{\frac{4CC_{UAV}}{\pi}} \,, \tag{2}$$

where *cd* is the tree crown diameter (cm) and $CC_{UAV}$ is the tree crown area.

Only trees with 50% of their crown surface within the plots were taken into account in sampling. An exhaustive manual revision of the detected trees was done to avoid committing omission and commission errors. In order to ensure the quality of UAV-based measurements, all plots were reviewed by a different expert than the one who carried out the semiautomatic process of the sample plot measuring.

### 2.5. Allometric Equation

AGB for individual trees was estimated using an allometric equation developed specifically for the project (Table 4) [87]. A specific equation was developed for *Rhizophora* based on a destructive sample of 71 trees from the study area. AGB was defined as the sum of stem, stilt roots, branches, leaves, and fruits biomass. The AGB allometric equation adjusted in the project has considered two independent variables (tree crown diameter and total height). These variables were the necessary ones for the estimation of tree AGB based on the photogrammetric information. The point clouds obtained during the UAV-based sampling of this study provided information of crown diameter and tree height at the individual tree-level. Based on the single-tree estimates of AGB, this attribute was computed for each field plot (Table 5).

**Table 4.** Tree allometric equation used for aboveground biomass estimates.

| Equation | $r^2$ | Number of Individuals | CD Range (cm) | *h* Range (cm) |
|---|---|---|---|---|
| $agb = 0.004932696 \times cd^{1.9869} \times h^{0.7166}$ | 0.93 | 71 | 10.5–210.0 | 37–285 |

*agb* = estimated individual tree oven-dry aboveground biomass (g); *h* = total height (cm); *cd* = tree crown diameter (cm).

**Table 5.** Summary of AGB results for the 95 UAV-based sample plots (m$^3$ ha$^{-1}$).

| Stratum | Number of Plots | Minimum | Mean | Maximum | Standard. Deviation |
|---|---|---|---|---|---|
| I | 55 | 0.00 | 0.33 | 1.60 | 0.49 |
| II | 42 | 0.00 | 8.05 | 36.93 | 9.72 |

### 2.6. Aboveground Biomass Modelling and Performance Assessment

Remote sensing data from Sentinel-1 and Sentinel-2 were used to enhance estimators of AGB predictions under the model-assisted inferential framework since this study was designed according to design-based principles. This method requires of models that relate AGB to the variables extracted from satellite data.

In the current study, SVR was used to estimate machine-learning models of the mean function. SVR has been used with good results in other remote sensing derived biomass estimations including mangrove plantations [29,30,37,88,89]. The SVR basis is to transform the multidimensional regression problem into a linear one to predict one-dimensional variables. This problem is solved by using appropriate kernel functions to map the training data into a new hyperspace feature [90]. In this study, the radial basis function (RBF) kernel was used due to its wide usage in other studies for modelling forest AGB [29,30,91]. The Vapnik's $\varepsilon$-insensitive loss function [92] was used to reduce model complexity by ignoring differences between predicted and true values smaller than $\varepsilon$. In order to minimize problems due to overfitting and achieve parsimonious models the best kernel-parameter combination of $\varepsilon$, the regularization parameter ($C$) and the kernel width ($\gamma$) was selected using the grid search method. During this step a 10-fold cross-validation was performed to assess the accuracy of the models.

To improve the accurateness of the models a backwards selection of predictors based on the predictor importance ranking was used by applying the recursive feature elimination (RFE) and relative variable importance algorithms in R (*caret* package) [93]. The relative variable importance was also assessed using the same R-package.

Three different models were adjusted; one per each data source and another one by combining both satellite datasets. A selection of predictive variables was applied in a first step except for the Sentinel-1 model. In order to assess the goodness-of-fit of the models, predictions were compared to the validation dataset using a variety of metrics: absolute (RMSE), mean absolute error (MAE) and coefficient of determination ($r^2$):

$$\text{RMSE} = \sqrt{\frac{\sum_1^n (\hat{y}_i - y_i)^2}{n}} \, , \tag{3}$$

$$\text{MAE} = \frac{1}{n} \sum_1^n |\hat{y}_i - y_i| \, , \tag{4}$$

$$r^2 = 1 - \frac{\sum_1^n (y_i - \hat{y}_i)^2}{\sum_1^n (y_i - \overline{y})^2} \, , \tag{5}$$

where $n$ is the total number of validation plots; $y_i$ is the observed AGB value plot i; $\hat{y}_i$ is the predicted AGB value for plot i, and $\overline{y}$ is the mean of observed AGB values for all validation sample plots.

Akaike information criterion (AIC) was used to compare the performance of the different models. AIC has been recently used for comparing models in other studies that have estimated AGB based on remotely-sensed data [38,94].

*2.7. Aboveground Biomass Estimation Methods*

The fitted SVR regression models were used to estimate AGB of the entire area by a model-assisted procedure. The design used allowed to estimate AGB on the basis of stratum and region-specific information. The plantation area was tessellated into grid cells using regular grids with the same area as backscatter image raster (400 m$^2$). Utilizing the nomenclature proposed by Särndal, Swensson, and Wretman [95] used in Næsset et al. [24] for a stratified random sampling (STRS), the entire population of grid cells in the study area is named $U$, where $U = \{1, ..., N\}$. Let $U$ be partitioned into $H$ non-overlapping strata, $U_h$. In this case $H = 2$. Let $N_h$ denote the size of $U$, with $h = 1, \ldots, H$. Let $b_k$ be the AGB of the $k$:th unit in the population.

The following estimator was used to estimate the mean AGB from the purely UAV-based sampling for each stratum:

$$\hat{B}_{STRSh} = \frac{\sum_{k \in s_h} b_k}{n_h} \, , \tag{6}$$

where $s_h$ is a sample of fixed size $n_h$ randomly designed.

The mean AGB for a particular stratum may be estimated using the model-assisted regression estimator (MAR) described in Næsset et al. [26] as follows:

$$\hat{B}_{MARh} = \frac{\sum_{k \in U_h} \hat{b}_k}{N_h} + \frac{\sum_{k \in s_h} \hat{e}_k}{n_h} , \tag{7}$$

where $\hat{b}_k$ is predicted AGB for the $k$:th grid cell, $N_h$ is the total number of grid cells for the stratum $h$, and $\hat{e}_k = b_k - \hat{b}_k$. The first term of Equation (7) is the synthetic regression estimator described in Särndal, Swensson, and Wretman [68]. This estimator is a sum of model estimates of each element in the population.

On the other hand, the second term is a Horvitz-Thompson estimator of the bias between the model predictions and the observed values in the sample for the stratum $h$. The Horvitz-Thompson estimator functions as a correction factor that makes the MAR asymptotically unbiased when $n_h$ is not too small [95].

The following estimator of the variance of the mean AGB estimation from the UAV-based alone was used:

$$\hat{V}(\hat{B}_{STRSh}) = \frac{\sum_{k \in s_h} (b_k - \hat{B}_{STRSh})^2}{n_h(n_h - 1)} \tag{8}$$

The variance of the mean AGB for the MAR was estimated as follows:

$$\hat{V}(\hat{B}_{MARh}) = \frac{\sum_{k \in s_h} \left( \hat{e}_k - \frac{\sum_{k \in s_h} \hat{e}_k}{n_h} \right)^2}{n_h(n_h - 1)} \tag{9}$$

The stratified estimator was used to estimate mean AGB for the entire study area. Equation (10) is the stratified estimator of AGB for the UAV-based sample and Equation (11) is the stratified estimator of mean AGB for the MAR:

$$\hat{B}_{STRS} = \sum_h \frac{N_h}{N} \hat{B}_{STRSh} , \tag{10}$$

$$\hat{B}_{MAR} = \sum_h \frac{N_h}{N} \hat{B}_{MARh} \tag{11}$$

Finally, the following variance estimators for the entire study area were used:

$$\hat{V}(\hat{B}_{STRS}) = \sum_h \left( \frac{N_h}{N} \right)^2 \hat{V}(\hat{B}_{STRSh}) , \tag{12}$$

$$\hat{V}(\hat{B}_{MAR}) = \sum_h \left( \frac{N_h}{N} \right)^2 \hat{V}(\hat{B}_{MARh}) \tag{13}$$

The standard errors (SE) were calculated for each stratum as the square root of the estimator of the variance of the mean AGB based on UAV-based and model-assisted methods, respectively. The relative efficiency (RE) parameter was computed for each model-assisted to compare UAV-based sample precision with the different model-assisted precision as follows:

$$RE = \frac{\hat{V}(\hat{B}_{STRS})}{\hat{V}(\hat{B}_{MAR})} , \tag{14}$$

where RE is the relative efficiency of different model-assisted over purely UAV-based sample. The greater than 1.0 is RE the higher is the efficiency of model-assisted estimates than UAV-based and the larger is the UAV-based sample size required to achieve the same results as the model-assisted method.

## 3. Results

### 3.1. Tree Measurements

Figure 3 shows the results of location and measurement of individual trees from the semiautomatic processing of the photogrammetric point clouds manually revised.

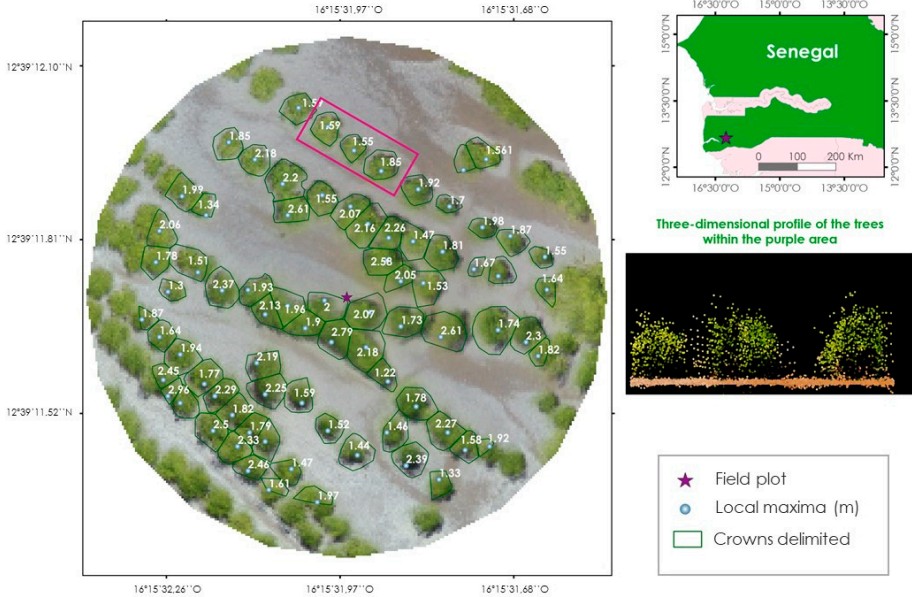

**Figure 3.** Example of individual tree detection from UAV-derived CHM, local maxima (m), and crown delineation for a sample plot.

The correlation coefficient between field-measured heights and UAV-measured heights was significant with a value of 0.95 (intercept of 6.1 cm and slope of 1.07) (Figure 4a). Paired *t*-test showed that both measures are significantly different. The mean difference between field height and the UAV height was 11.59 cm (95% confidence interval from 7.99 cm to 15.19 cm). UAV point cloud measurements tended to underestimate the tree heights (Table 6). The root mean square error (RMSE) for individual tree heights was 0.21 m.

**Table 6.** Summary of the measured and estimated tree variables (m).

|  | Field Tree Height | UAV Tree Height | Field Tree Crown Diameter | UAV Tree Crown Diameter |
|---|---|---|---|---|
| Minimum | 0.35 | 0.23 | 0.08 | 0.01 |
| Mean | 1.12 | 1.00 | 0.85 | 0.81 |
| Maximum | 3.40 | 2.89 | 3.03 | 2.67 |

Figure 4b shows the strong linear relationship between the point cloud-derived and field measured tree crown diameters ($r^2 = 0.75$). As in the case of tree height, tree crown diameters were slightly underestimated with a bias of 3.17 cm (95% confidence interval from $-3.47$ cm to 9.81 cm). The two-sided *t*-test revealed that there were no significant differences ($p \geq 0.95$) between the mean of the crown diameters measured over the point clouds and reference values. The RMSE of tree crown measurements was 0.32 m.

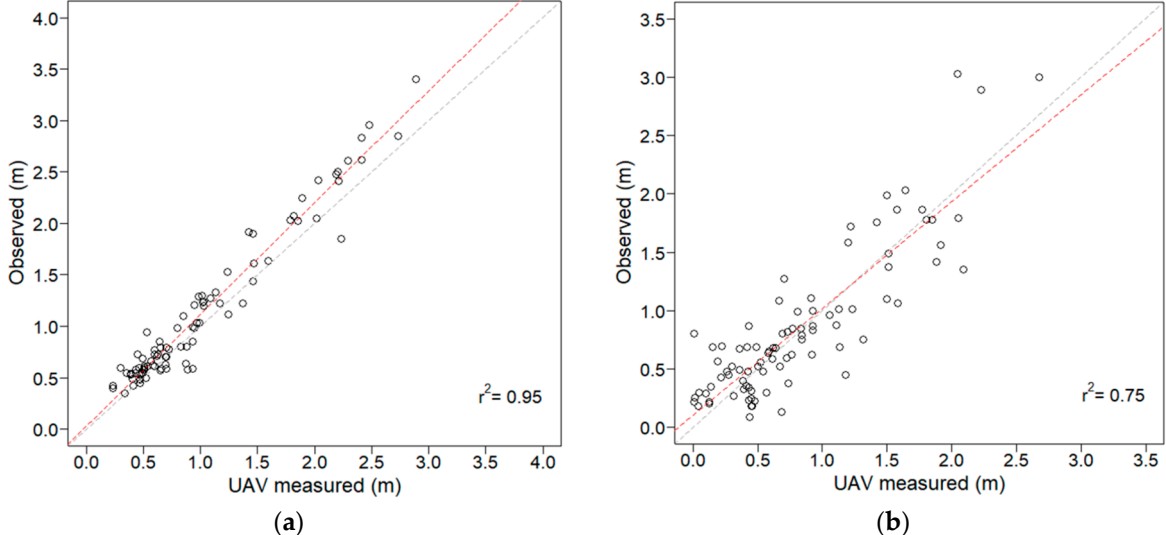

**Figure 4.** Scatter plot detailing the coefficient of determination ($r^2$) between (**a**) field measured height (m) and the maximum height from UAV-derived point clouds for individual trees and (**b**) field measured tree crown diameters (m) and tree crown diameters from UAV-derived point clouds. The red line shows the linear fit of the UAV-derived point clouds measurements and field observed values. The grey line in the center indicates 1:1.

*3.2. Model Fitting*

AGB in the 95 UAV-measured plots was regressed against the predictor variables computed from Sentinel-1 and Sentinel-2 data using SVR. Table 7 shows the performance for the validation dataset of the different SVR models generated using Sentinel-1, Sentinel-2 and the combination of both datasets to estimate the mangrove plantations AGB (Mg ha$^{-1}$). The selected SVR models explained 71–90% of the variability. AGB modelling results showed higher accuracy using SAR data than using optical data alone or in combination with SAR data. The SVR models for AGB contained a maximum of five explanatory variables (Table 7) with Sentinel-2-derived vegetation indices being more important than spectral bands. SAR-based model achieved the highest $r^2$ and the smallest RMSE and MAE values, while the model based only on spectral indices showed the worst results. The combination of both satellite datasets did not improve the RMSE and MAE as the results are slightly lower compared to the SAR-based model. Regarding AIC values, the best model for AGB included both SAR and multispectral data. Scatter plots of observed versus predicted AGB in the validation dataset for the different models are displayed in Figure 5. Figure 6 summarizes the variable importance metrics for AGB model predictors.

**Table 7.** Performance of the selected SVR models.

| Inputs | Selected Variables | $r^2$ | RMSE (Mg ha$^{-1}$) | MAE (Mg ha$^{-1}$) | AIC |
|---|---|---|---|---|---|
| Sentinel-1 | VH | 0.90 | 2.22 | 0.89 | 89.27 |
| Sentinel-2 | PSSRa, NDVI2, GNDVI, IRECI, OSAVI | 0.71 | 3.74 | 1.91 | 218.23 |
| Sentinel-1 + Sentinel-2 | VH, IRECI, SAVI, OSAVI | 0.89 | 2.35 | 1.20 | 67.33 |

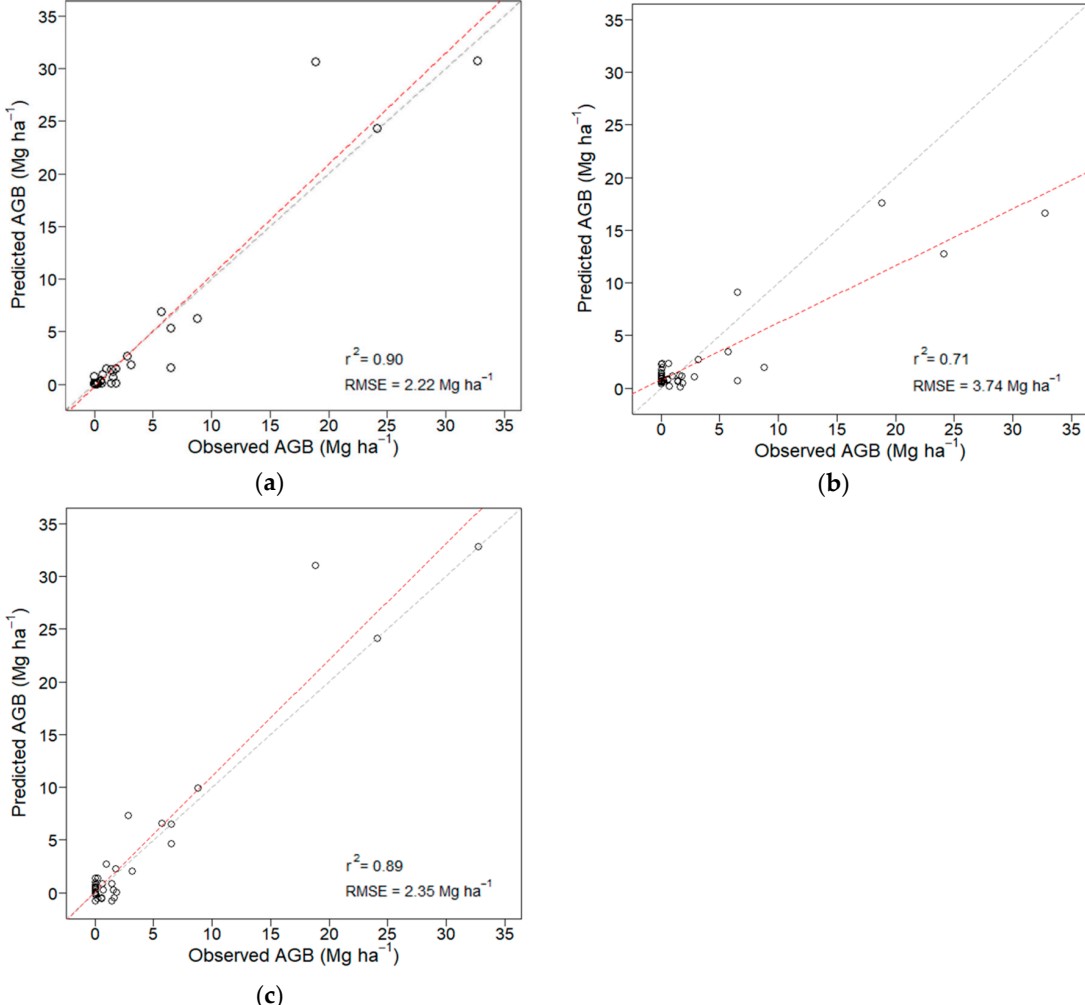

**Figure 5.** Scatterplot of observed against predicted values from the cross-validation for the (**a**) Sentinel-1 SVR model, (**b**) Sentinel-2 SVR model, and (**c**) Sentinel-1 + Sentinel-2 SVR model. The red line shows the linear fit of the predicted and observed values. The grey line in the center indicates 1:1.

*3.3. Estimations of Aboveground Biomass*

For the entire studied area, the UAV-based estimation produced a mean AGB value of 2.90 Mg ha$^{-1}$ (SE = 0.55 Mg ha$^{-1}$) (Table 8). Using remotely-sensed auxiliary data under a model-assisted framework the corresponding estimations ranged between 2.51 Mg ha$^{-1}$ (SE = 0.43 Mg ha$^{-1}$) to 3.66 (SE = 0.38 Mg ha$^{-1}$). The combination of SAR and optical data showed the greatest RE. Consequently, combining both spaceborne data resulted in an improvement in an efficiency improvement of 115% compared to the purely UAV-based method.

**Table 8.** Estimated mean AGB ($\hat{B}$) and standard error (SE) estimates (Mg ha$^{-1}$) based on UAV-based sampling and model-assisted estimation from Sentinel-1, Sentinel-2, and the combination of both satellite data.

| Stratum | UAV-Based | | Model-Assisted | | | | | | | | |
|---|---|---|---|---|---|---|---|---|---|---|---|
| | | | Sentinel-1 | | | Sentinel-2 | | | Sentinel-1 + Sentinel-2 | | |
| | $\hat{B}$ | SE | $\hat{B}$ | SE | RE | $\hat{B}$ | SE | RE | $\hat{B}$ | SE | RE |
| I | 0.33 | 0.35 | 0.99 | 0.31 | 1.27 | 0.75 | 0.30 | 1.32 | 0.95 | 0.35 | 0.98 |
| II | 8.05 | 1.50 | 8.50 | 0.97 | 2.37 | 6.04 | 1.16 | 1.68 | 9.12 | 0.88 | 2.87 |
| All | 2.90 | 0.55 | 3.49 | 0.38 | 2.06 | 2.51 | 0.43 | 1.61 | 3.66 | 0.38 | 2.15 |

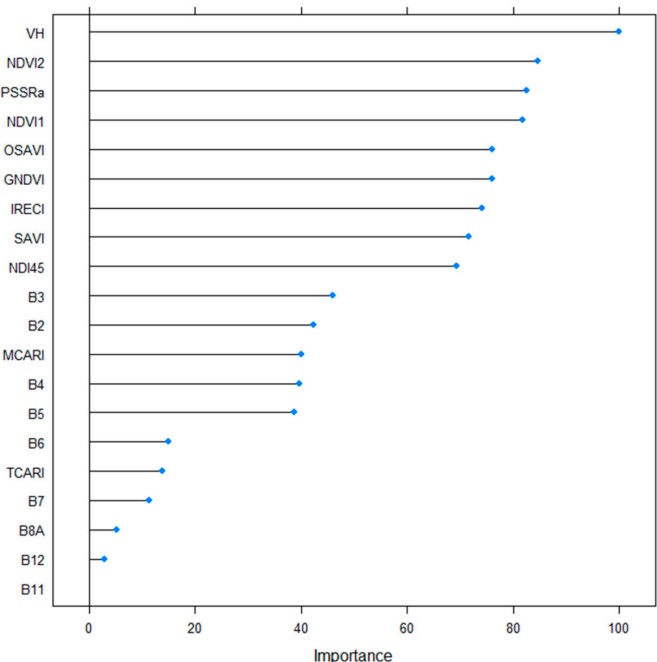

**Figure 6.** Variable importance measures generated for a SVR model including all variables.

The mean model-assisted AGB estimates in the stratum I ranged from 0.75 to 0.99 Mg ha$^{-1}$ (SE = 0.30–0.35 Mg ha$^{-1}$) while for the UAV-based the mean AGB estimate was 0.33 Mg ha$^{-1}$ (SE = 0.35 Mg ha$^{-1}$). The highest RE was obtained using Sentinel-2-assisted estimates (RE = 1.32). For Stratum II, the estimated mean AGB values ranged between 6.04 and 9.12 (SE = 0.88–1.16 Mg ha$^{-1}$). The best results in terms of RE were achieved using the combination of spectral indices from Sentinel-2 and the backscatter from Sentinel-1 (RE = 2.87). Figure 7 shows the estimated AGB maps generated using the three adjusted SVR models. All maps showed similar patterns of AGB density distribution, but the model based on Sentinel-2 data variables led to lower AGB estimated values than the models including Sentinel-1 VH polarization.

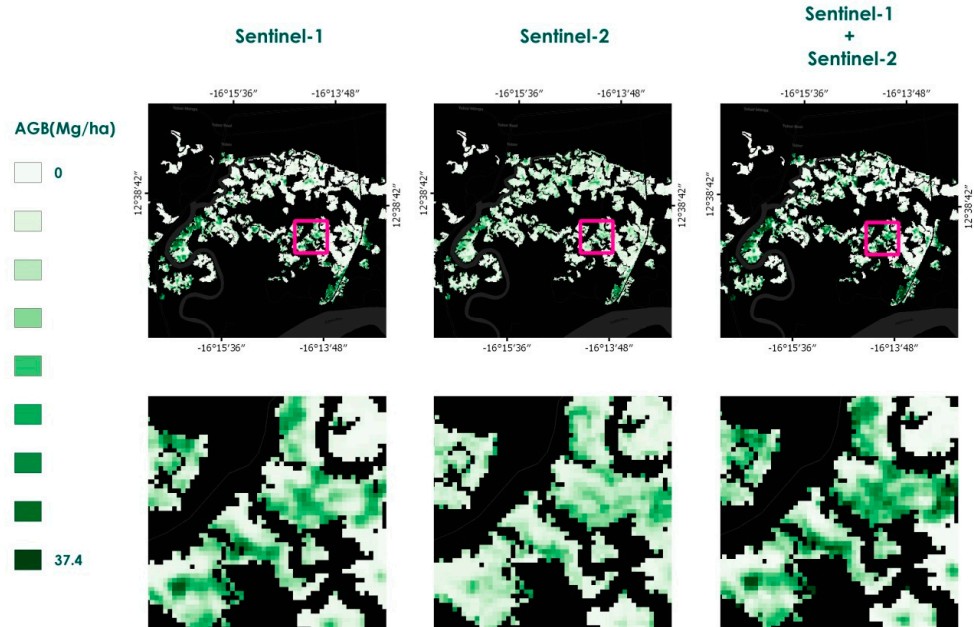

**Figure 7.** Study area AGB maps derived from the three SVR models used in this research. The upper row shows a general view of the AGB estimations while the lower row shows details at a smaller scale.

## 4. Discussion

This study described a new method for large-scale forest AGB monitoring in remote and difficult-to-work areas by combining the use of UAVs for aerial plot measurements and data from Sentinel-1 and Sentinel-2 as auxiliary information. This methodology was used to assist in the development of monitoring of large-scale carbon sequestration projects on multiple plots. In the last few years, CHMs derived from UAV imagery have been extensively used to determine tree locations for various purposes, such as measuring tree height and crown sizes, estimating diameter at breast height, and assessing AGB [48,54,57,61,96]. In this case, however, an affordable UAV has been used to generate a specific CHM for each sample plot to measure AGB of plots. Sentinel imagery was used as auxiliary data to estimate AGB in the different strata and the entire study area in a model-assisted framework, as in Næsset et al. [26], since the estimators are approximately design-unbiased [32].

Although high tide can make the generation of a quality DIM-derived DTM difficult or even impossible, all flights were planned to avoid this condition. In addition, flat terrain and open canopies helped with the DTM generation, as in [97,98]. Successful CHMs were built in every aerial plot. ITC delineation algorithms used in the sample plot measurement led to a more efficient, accurate, and productive job. Sample plots were always placed at the central zone of each point cloud which had the highest overlaps.

The accuracy of tree height measures lower tree heights for our study sites and the values presented in previous studies [54,57,99]. This fact can be due to the significantly lower height that present those trees in comparison with previous studies. However, RMSE (20.64 cm) is considered relatively small, so results on measurements of tree heights were satisfactory. UAV-based tree height measures were negative-biased, meaning that this parameter was underestimated in agreement with other surveys [51,65,100]. The comparison between UAV and ground measurements showed that it is possible to make conservative and realistic measures of tree heights from the photogrammetric 3D models. The findings from our study demonstrated that UAV-derived point clouds may be successfully used to estimate tree heights and crown diameters in recently established mangrove reforestations. From this information, we can accurately estimate AGB if accurate allometric equations based on tree height or crown dimensions are available. For this study, conservative values of AGB were estimated for each sample plot resulting from the negative biased estimations of individual tree heights. A conservative approach in the estimation of AGB is recommended by the different forest carbon standards to avoid overestimation of GHG removals. A potential weakness of the methodology followed in assessing the sample plot AGB by measuring tree variables over UAV-derived point clouds is the lack of accuracy of ITC algorithm with increasing density of stands [48,101]. Tree crown diameter measurements may be inaccurate in mature forests due to irregular shapes of crowns [48]. In addition, this method may not be suitable in forests with heterogeneous structure since only the upper canopy is detected [61].

While accuracy of UAV-based tree variables measurement was studied using a 100-tree validation sample, the accuracy of plot level AGB estimations was not investigated. Sample plot AGB prediction uncertainty may be assessed using a Monte Carlo bootstrapping approach [102]. We recommend that future research focus on assessing UAV-based AGB estimation uncertainty at the plot level in order to evaluate all potential sources of error.

The second phase of this study was to generate precise estimations of AGB for the whole study area combining the stratified systematic sample of UAV-based plots and full-coverage moderate resolution SAR and optical satellite data. As expected, the models fitted using SAR auxiliary data were able to explain more variability ($r^2$ = 0.89–0.90) and had lower RMSE values (RMSE = 2.22–2.35 Mg ha$^{-1}$) than the model based only on optical imagery ($r^2$ = 0.71, RMSE = 3.74 Mg ha$^{-1}$). These results are satisfactory compared to previous studies on the integration of SAR and optical data for AGB estimation. Aslan et al. [103] estimated AGB of the coastal wetland vegetation in Indonesian Papua by fusing Landsat-8 OLI and ALOS-2 PALSAR-2 data with $r^2$ = 0.46. Pham et al. [30] reported a $r^2$ value of 0.60 in a mangrove plantation on the northern coast of Vietnam. They used SVR models

with Sentinel-2 and ALOS-2 PALSAR-2 data. Jachowski et al. [89] used very high resolution GeoEye-1 and ASTER GDEM V2 elevation data (resolution of 30 m) in mangroves of Southwest Thailand with $r^2$ = 0.66. Higher goodness of fit found in this study may be due to low AGB density in the plantation area compared to the rest of studies which were developed in mangrove forests with denser biomass.

Results showed that VH backscatter was the most important variable for modelling AGB in the study area. This result was consistent with the findings of Alan et al. [104] and Pham et al. [30] Although VH polarization showed the best performance estimating AGB in the study area, not using other C-band polarization was the main limitation. Other studies found strong correlation between VV, HH, HV, VV/HH, HH/HV, or VV/HV and AGB [30,37,104]. The other main limitation could be the saturation of C-band in high biomass areas. While C-band is not sensitive to values of AGB exceeding 50–70 Mg ha$^{-1}$, the saturation level of the AGB estimation in mangrove forests using L-band has been detected at 100–150 Mg ha$^{-1}$ [25,28,30,105,106]. Nevertheless, sample plot AGB in our study area ranged from 0 to 36.93 Mg ha$^{-1}$ and the C-band is favored for these low biomass areas, i.e., forest regeneration or young plantations [22]. Moreover, Prior studies using machine learning methods and L-band showed overestimations of AGB in mangrove plantations at values bellow 50 Mg ha$^{-1}$ [30,38]. This study showed that VH polarization from Sentinel-1 may be used to correctly estimate AGB in mangrove plantations bellow 30 Mg ha$^{-1}$ using machine learning algorithms, such as support vector machine.

A model combining Sentinel-1 and Sentinel-2 was adjusted to enhance the sensitivity of C-band backscatter at high AGB levels in mangroves (i.e., areas with pre-existing trees). Although this model resulted in a decrease of accuracy for the very low AGB densities stratum, the findings showed that it was more sensitive to higher AGB levels. No saturation issues were found in the study area for Sentinel-1 data since sample plot AGB values were bellow typical saturation levels for mangrove areas. In this sense, more research is needed to analyze whether Sentinel-2 multispectral data is able to minimize the saturation problem in denser AGB level sites, such as natural mangrove forests or older plantations.

Indices using NIR spectrum (B8 and B8A) and red edge bands were the most important variables among optical data for predicting AGB (Figure 6). This is consistent with the findings of Sibanda, Mutanga, and Rouget [107]. None of the individual bands were included in the models. Unlikely our results, NIR bands had the most important role rather than vegetation indices in previous studies [30,89]. However, Alan et al. [104] found the highest correlation values for spectral indices, such as IRECI, for modelling AGB in mangrove forests.

The findings of this study have shown that incorporating Sentinel data in a young mangrove plantation monitoring may enhance AGB estimations and achieve more accurate results compared to those obtained by a UAV-based only inventory. Combining data from the stratified systematic sampling and a model based on data from the Sentinel constellation reduced, in all cases, the SE values except for Stratum I model-assisted estimation of AGB for the low-density biomass stratum using auxiliary data from both satellites produced the greatest SE value (SE = 0.35 Mg ha$^{-1}$). However, this model was the most sensitive to higher biomass densities. The best RE values were achieved in Stratum II using the combination of both satellite data. For this stratum and model, 187% more plots would be needed to obtain the same accuracy by a purely UAV-based inventory under simple random sampling. In this way, a hybrid approach might be recommended, i.e., using Sentinel-1 variables as auxiliary data for the very low density stands (plantation areas in the first years of establishment) and a combination of SAR and multispectral data for older stands or areas with pre-existing trees.

A correction of the bias was used in this study by a Horvitz-Thompson estimator under the model-assisted framework. This conferred an advantage on the estimation of AGB as compared to purely model-based inference because mean and total unbiased estimators are not model accuracy dependent, as the model is only improving the design-based estimator [32,95].

The results of our study have demonstrated that Sentinel-1 and Sentinel-2 data may be used to develop accurate, rapid, up-to-date estimates of AGB of young mangrove plantations in large areas. The European Commission has adopted a free, full, and open data policy for all Copernicus data, so Sentinel products may be used as cost-effective data to reduce the number of sample plots and improve the results of afforestation, reforestation, and/or revegetation mangrove plantation monitoring. The use of multi-source remote sensing data helped in the stratification phase and led to better results in the modelling and estimation phases. In addition, using UAVs made sample plot measuring easier, reduced costs of inventory, and made it possible to work in the most remote areas. However, the promising results reported in this research must be tested in older plantations and natural forests where saturation issues are expected and denser canopy cover makes ITC delineation more difficult.

## 5. Conclusions

The main innovation of this study was the development of a novel approach to large-scale monitoring of young mangrove forests that combine spaceborne optical and radar data. This approach has enhanced traditional field plot measurements with semiautomatic measures using low-cost UAV-derived photogrammetric point clouds. The assessment of accuracy showed that individual tree variables were successfully measured and confirmed that the followed workflow may be an alternative for quickly gathering precise measurements from sampling plots in remote areas.

This study has also confirmed the good performance of SVR modelling AGB in mangrove plantations. Although the Sentinel-1-based SVR model had the best results in terms of $r^2$, RMSE, and MAE, the integration of Sentinel-1 and Sentinel-2 data led to achieve more accurate estimations in the higher biomass areas. The study has demonstrated that remote sensing-assisted monitoring substantially improved the precision of AGB estimates compared to pure UAV-based inventory. The integration of radar and optical data produced the lowest standard errors of the model-assisted estimations, more especially in the higher AGB stratum. Specific studies focusing on the shape and tree size influence on the UAV-derived accuracy measures and the remote sensing-based models are needed to support mangrove AGB monitoring in different conditions depending of country landscapes.

**Author Contributions:** Conceptualization: J.A.N., N.A., and A.F.-L.; data curation: J.A.N., N.A., A.F.-L., J.E., and M.L.G.-C.; formal analysis: J.A.N., N.A., A.F.-L., J.E., and M.L.G.-C.; funding acquisition: P.R.-N.; methodology: J.A.N., N.A., and A.F.-L.; project administration: J.A.N. and P.R.-N.; software: J.A.N., A.F.-L., and J.E.; visualization: J.A.N. and J.E.; writing—original draft: J.A.N.; writing—review and editing: J.A.N. and J.E.

**Funding:** This research was funded by the Fonds Français pour l'Environnement Mondial (FFEM) through its partnership with Livelihoods Venture, the exclusive advisor of the Livelihoods (Carbon) Fund. José Antonio Navarro's participation was also supported by a predoctoral grant [DI-15-08093] and Nur Algeet [PTQ-14-07206] and Mariluz Guillén-Climent [PTQ-12-05748] by postdoctoral grants awarded by the 'National Programme for the Promotion of Talent and Its Employability' of the Ministry of Economy, Industry, and Competitiveness (Torres-Quevedo program), which are partially funded by the European Social Fund (ESF) from the European Commission.

**Acknowledgments:** We are grateful to the Senegalese NGO Oceanium, especially to Octavio Fleury for the support during the UAV data collection.

**Conflicts of Interest:** The authors declare no conflict of interest. The funders had no role in the design of the study; in the collection, analyses, or interpretation of data; in the writing of the manuscript, or in the decision to publish the results.

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
