# Peer review of "Integration of UAV, Sentinel-1, and Sentinel-2 Data for Mangrove Plantation Aboveground Biomass Monitoring in Senegal"

_remotesensing, doi:10.3390/rs11010077_

Round 1
Reviewer 1 Report
This is a nice novel application of drones for measuring AGB in Mangroves which I think has relevance to the community. I was particularly pleased to see that a very affordable drone was used, this makes the method more accessible to readers.
I would have liked to see more in the discussion about how applicable the method is to natural forests, and a few minor changes in the methods and results - the comments are annoted in the manuscript.

Author Response
Thanks for your interest in our work and for your helpful comments that will greatly improve the manuscript. We have tried to do our best to respond to the points raised. We appreciate the opportunity to clarify the points that you have brought up. All comments have been answered in the attached document.

Reviewer 2 Report
Overall, this paper presents an interesting topic dealing with mangrove AGB estimation using multi-source remote sensing data. However, the current version has lots of shortcomings and weaknesses make it unsuitable for consideration for possible publication in Remote sensing. I found some sentences repetition issue and it must be rewritten. Seriously, the methods used in this paper is unclear at all. Authors must present a research framework how can they employ multi-source remote sensing data in the current work. Additionally, I did not see any maps presented mangrove AGB in the current version.
Specific comments:
- Introduction:
This part is lack of review on the methods for mangrove AGB estimation including parametric and non-parametric approaches.
- Materials & Methods:
+ Lines 128-133: Please add references. Please add more information on geographic and climate conditions with proper references
Fig.1 should modify to make it easy and clear to see. The UAV training and validation datasets should be added.
+ Authors should add a table describing the remote sensing data acquisition.
+ What is the ground condition during the field survey?
+ Line 152: Why only VH was used in this study as some recent studies pointed out the multiple-PolSAR play an important role in estimating mangrove AGB.
+ Sentinel 1 data acquisition on 18th July 2017, from 11th August 2017 and from 28th September 2017 while Sentinel 2 data on 24th Feb and 9 the March. And UAV was achieved in July and Aug 2017. How the authors compute data fusion? It must be explained in the revisions.
+ It is not clear how the authors combine multi-feature learning derived from Sentinel-1, 2 and UAV?
+ Lines 183-188: It is unclear what is the purposes for the use of RF for classifying Sentinel-2 data? Did authors want to apply for generating a mangrove map then applying the SVR for estimating AGB?
+ A flowchart describes an entire process for estimating mangrove AGB must be added and clearly explaned.
+ A plot size is about 314.2 m2 while a pixel size of the Sentinel-2A is 10 m. How did the authors deal with this issue?
+ It is unclear how the author test and validate the results? How many plots used for training and validation datasets among 95 UAV-measured plots?
- Results
+ Variable importance must be tested
+ How about the data saturation for the use of SAR, optical, and multi-source data? Authors must show the scatter plots between measured plots and each input variable?
+ AGB maps for different periods must be showed in the manuscript
+ Table 6. are they statistically significant difference? Authors should use the AIC to test. See paper entitled :
Biomass estimation of Sonneratia caseolaris (l.) Engler at a coastal area of Hai Phong city "(Vietnam) using ALOS-2 PALSAR imagery and GIS-based multi-layer perceptron neural networks" for more details.
+ Fig. 4: Authors should add R2 and RMSE in the scatterplots.
- Discussion: this chapter is very weak. Authors did not mention the current limitations. And authors did not make enough comparison between this study and other similar studies in different case studies for mangrove AGB estimation.
+ Data saturation should be discussed more thoughtfully. See paper above for more detail
+ What are the benefits for the use of multi-source remote sensing data for mangrove AGB estimation?
Author Response
We would like to thank you for your time. We found your comments extremely helpful and have revised accordingly. In our opinion, the document has been significantly improved with your comments. We have worked on the shortcomings and weaknesses, enhancing each manuscript section and providing the resources to frame our study. AGB maps has been included in the manuscript.

Reviewer 3 Report
The manuscript presents a method for estimating aboveground biomass of mangrove plantations in Senegal using UAV photogrammetric data, Sentinel-1 C-band SAR and Sentinel-2 multispectral optical data at the scale of small plots and at larger (landscape) scale. The proposed method represents an interesting option for carbon assessment in (young) mangroves and, in general, I think the paper could be published in Remote Sensing. I have some comments with respect to the methodology, but nothing major. However, a real problem of the manuscript is the English writing. That needs to be improved substantially.
Additional comments:
1) If I understand correctly, in each of the plots, a single tree was measured (height, crown diameter). Biomass was not estimated based on field data at plot level. When in the manuscript it is referred to field-based estimates of biomass, that actually refers to the biomass estimates derived with the aid of UAV estimates of per tree height and crown diameter and the allometric Equation in Table 3. The terminology should be changed to make this clearer.
2) Also, we do not know how accurate the UAV derived biomass estimates at plot level are because only the biomass of a single tree in each plot is known, right? That should be addressed/discussed in the manuscript.
3) In section 2.6, it is stated that a probability-based design is presumed for inference; in section 2.7, instead, it is stated that design-based inference principles were applied. Please clarify.
4) Three Sentinel-1 acquisitions were considered for estimating biomass. Experience shows that the sensitivity of C-band backscatter to biomass varies substantially dependent on the environmental imaging conditions and that integration of multi-temporal observations is crucial to achieve good retrieval results. Maybe I have missed it when reading the manuscript, but I didn’t see a description if only a single scene was selected for retrieving biomass or if all were considered. Would be interesting to see the differences between the different acquisition dates.
5) Avoid a discussion of saturation effects in SAR backscatter. The saturation limits mentioned in the Discussion section, in particular those for C-band, are not valid. C-band has been used for biomass retrieval in forests with biomass levels much higher than 30 t/ha. In this study very low biomass ranges are considered (<40 t/ha), so a discussion of saturation limits is not necessary.
6) The description of the pre-processing of S1 data is lacking detail. Was multi-looking applied, and if yes which multilooking factors. Why were the images processed to 20 m pixel size and later on oversampled to 10m pixel size? S1 GRDs are provided with a ground range pixel spacing of 10 m (i.e., oversampled in range).
7) The variance estimators in section 2.7 do not consider spatial correlation. I would expect spatial correlation to have a significant effect on the variance. Could you at least roughly quantify what the effect of correlation when deriving biomass estimates for the study area is?
Author Response
We thank the reviewer’s appreciation about the subject and methodology of our manuscript. Thank you for your comments and the opportunity to revise our paper. Bellow we have answered each one of your comments which have improved substantially our document. Furthermore, Language editing for the manuscript has been done. We have used the services of a professional editing company for the same.

Reviewer 4 Report
It is my opinion that the paper can be published.
Author Response
Response to Reviewer 4 Comments
Point 1: It is my opinion that the paper can be published.

Response 1: Thank you very much
Round 2
Reviewer 2 Report
Dear authors;
Thanks for the revised version. The authors have addressed all of my comments; therefore, the quality of this manuscript has improved compared to the previous version. However, some minor points should be taken into account before it can be accepted for publication in Remote Sensing.
- Fig. 1: The quality of this figure is low. and can not be clearly seen. Authors should change the color and improve the spatial resolution. For the figure, the minimum spatial resolution should be at least 600dpi.
- Fig. 2: The quality is too low. Please improve the spatial resolution.
- Line 290-297. Please add the reference which allometric equation used for AGB estimation.
Did author consider the dry weight of roots of Rhizophora mangle? I believe the total AGB of Rhizophora mangle tree should be counted for its roots (stilt) based on DBH.
- If not, authors have to highlight the limitations of the current work.
Author Response
- Fig. 1: The quality of this figure is low. and can not be clearly seen. Authors should change the color and improve the spatial resolution. For the figure, the minimum spatial resolution should be at least 600dpi.
Colors of sampling have been changed and spatial resolution has been improved to 600 dpi. Thank you for your assessment.
- Fig. 2: The quality is too low. Please improve the spatial resolution.
Done. Colors have been also changed to improve the quality.
- Line 290-297. Please add the reference which allometric equation used for AGB estimation.
Done
Did author consider the dry weight of roots of Rhizophora mangle? I believe the total AGB of Rhizophora mangle tree should be counted for its roots (stilt) based on DBH.
- If not, authors have to highlight the limitations of the current work.
Thanks to your comment, it was taken into consideration. We have included it in the manuscript.
Reviewer 3 Report
I have no further comments
Author Response
Thank you very much